# Seroprevalence of Poliovirus Types 1, 2, and 3 Among Children Aged 6–11 Months: Variations Across Survey Rounds in High-Risk Areas of Pakistan

**DOI:** 10.3390/vaccines13101067

**Published:** 2025-10-19

**Authors:** Imtiaz Hussain, Ahmad Khan, Muhammad Umer, Muhammad Sajid, Haider Abbas, Muhammad Masroor Alam, Altaf Bosan, Jeffrey Partridge, Rehan Hafiz, Anwar-ul Haq, Sajid Soofi

**Affiliations:** 1Centre of Excellence in Women and Child Health, The Aga Khan University, Karachi 74800, Pakistan; imtiaz.hussain@aku.edu (I.H.); ahmad.sherali@aku.edu (A.K.); muhammad.umer@aku.edu (M.U.); sajid.muhammad2@aku.edu (M.S.); haiderjamal12@gmail.com (H.A.); 2Department of Pediatrics & Child Health, The Aga Khan University, Karachi 74800, Pakistan; 3National Institute of Health, Islamabad 45500, Pakistan; ursmasroor@yahoo.com; 4Polio National Emergency Operations Center, Government of Pakistan, Islamabad 44000, Pakistan; ntfppei@eoc.gov.pk (A.B.); ncpei@eoc.gov.pk (A.-u.H.); 5Polio Program, Global Development, Bill & Melinda Gates Foundation, Seattle, WA 98109, USA; jeff.partridge@gatesfoundation.org (J.P.); rehan.hafiz@gatesfoundation.org (R.H.)

**Keywords:** poliovirus seroprevalence, population immunity, serotype immunity, serological surveys, children 6–11 months, Pakistan

## Abstract

**Background**: The current polio epidemiology in Pakistan poses a unique challenge for global eradication, with polio transmission dynamics influenced by regional variations in immunity and disparities in immunization coverage. This study assesses the immunity level for all three poliovirus types among children aged 6–11 months in polio high-risk regions of Pakistan. **Methods**: Four consecutive rounds of cross-sectional serological surveys were conducted in polio high-risk areas of Pakistan between November 2016 and October 2023. Twelve high-risk areas were covered in the first three rounds of the survey, while 44 high-risk areas were covered in the fourth round. 25 clusters from each geographical stratum were selected utilizing probability proportional to size. **Results**: Across the four rounds of the survey, 32,907 children aged 6–11 months from 2084 clusters and 32,371 households were covered. Comparative analysis across the survey rounds showed that seroprevalence of poliovirus type 1 was high in provinces (>95%), albeit consistently lower in Balochistan (going down to 89.7% in Round 4). Type 2 seroprevalence was significantly lower and more heterogeneous, from 34.6% in Sindh to 83.4% in Punjab, with sharp declines by round 4, particularly in Balochistan (40.4%). Type 3 seroprevalence was overall high (>94% in Punjab, Sindh, and KPK) but dropped in the last round, while Balochistan exhibited continually lower immunity (81.1%). **Conclusions**: The findings reflect the variations in population immunity to poliovirus in the country, with notable fluctuations over the years. The gaps in type 2 immunity over time and consistently lowest in Balochistan highlight the need for continued monitoring of immunity levels and adaptable vaccination strategies.

## 1. Introduction

Poliomyelitis (polio) has been one of the infectious diseases targeted for global eradication since 1988 by the World Health Assembly, following the successful eradication of smallpox in 1980. In the same year, the Assembly launched the Global Polio Eradication Initiative (GPEI) with the ambitious goal of eradicating poliovirus worldwide by the year 2000 [1]. Since then, remarkable progress has been made in reducing global polio cases by over 99.9%, and the successful elimination of two of the three strains of the wild poliovirus, WPV2 and WPV3, in 2015 and 2019, respectively [2,3,4]. Nonetheless, complete eradication remains elusive, as Pakistan and Afghanistan continue to report cases of wild poliovirus type 1 (WPV1) transmission even today [5,6]. The unfinished journey of polio eradication in Pakistan represents a complex interplay of scientific, socio-political, cultural, systemic, and logistical challenges that have hindered the achievement of a polio-free status [7].

The ongoing circulation of poliovirus in Pakistan is attributed to multiple factors, including vaccine hesitancy, mobile, migrant, and displaced population, geographical isolation, security concerns, and operational challenges in outreach immunization campaigns and immunization service provisions at fixed sites in hard-to-reach areas [8,9]. These challenges are more pronounced in high-risk areas that serve as persistent reservoirs for poliovirus circulation in the country. These areas include regions in Balochistan, Khyber Pakhtunkhwa (KP), Sindh, and densely populated urban centers like Karachi [10].

In Pakistan and other countries, poliovirus eradication efforts have been further complicated with the emergence of vaccine-derived polioviruses (VDPVs), particularly type 2 (cVDPV2) [11,12]. The emergence of cVDPV2 is associated with the trivalent oral poliovirus vaccine (tOPV), which comprises live-attenuated strains of poliovirus 1, 2, and 3. This vaccine has the potential to induce vaccine-associated paralytic poliomyelitis (VAPP) in individuals who receive the vaccine and those in close contact with them. VAPP happens when attenuated oral poliovirus vaccine sometimes relapses to neurovirulence, leading to paralysis that clinically resembles poliomyelitis caused by wild poliovirus. This OPV circulates in under-immunized populations, resulting in outbreaks of VDPV [3,4,13].

The global switch from tOPV to bivalent OPV (bOPV, containing only types 1 and 3) implemented in April 2016 was meant to eliminate the risk of VAPP and the emergence of cVDPV2 from the type 2 component of tOPV [14,15]. However, the switch created new challenges for maintaining population immunity against type 2 poliovirus, as children born after the switch did not receive type 2 OPV in routine vaccination unless they were vaccinated with monovalent type 2 OPV (mOPV2) in supplementary immunization activities (SIAs) or with inactivated polio vaccine (IPV) in SIAs and routine vaccination [3,16]. Sabin-like type 2, which was circulating before the withdrawal of tOPV, could continue to cause outbreaks of type 2 cVDPV among these children [17,18,19]. In 2024, Pakistan reported 74 cases of WPV1, and as of September 2025, 24 cases have been reported. For cVDPV2, no cases have been reported in the country since 2021 [20].

The immunization landscape in Pakistan has evolved significantly over the study period (2016–2023). The country has adopted several strategies to strengthen both routine immunization and supplementary immunization campaigns, in addition to the inclusion of IPV into the routine immunization schedule in 2015, the use of monovalent and bivalent OPV formulations in targeted campaigns, and the implementation of context-specific innovative approaches to reach children in high-risk and security-compromised areas [5,21]. These efforts have been supported by enhanced environmental surveillance systems to detect poliovirus circulation in sewage samples. This surveillance approach has proven particularly valuable in identifying virus transmission in the absence of paralytic cases [22,23].

Like in other countries, the COVID-19 pandemic disrupted the polio eradication efforts in Pakistan [24]. Polio vaccination campaigns were temporarily suspended in early 2020, leading to an estimated 40 million children missing polio vaccinations during this period [25]. Although campaigns resumed later, the disruption likely created new immunity gaps, potentially reversing some of the progress made in previous years [26,27].

Moreover, regional disparities in healthcare access, socioeconomic status, and security conditions contribute to heterogeneity in immunization coverage and population-level immunity in the country [28]. Urban slums, hard-to-reach areas, and conflict-affected regions often exhibit low vaccination coverage and high susceptibility to poliovirus circulation [29]. The migrant and mobile population within the country and across its borders, particularly with Afghanistan, is one of the major challenges in a populous country like Pakistan that has been impacting the polio eradication efforts on a sustained basis [30]. Cross-border movement between Pakistan and Afghanistan has been an uninterrupted path for poliovirus transmission in both countries, attested by genetic sequencing of isolated viruses transmitted [31,32]. Internal migration, including seasonal movement patterns and displacement due to natural disasters or conflict, also transmits poliovirus into previously polio-free areas, creating risk for the communities to maintain higher population-level immunity [33,34].

Understanding the immunological landscape across the diverse polio-high-risk regions is essential to understand geographic patterns of immunity, identify population vulnerability, and tailor interventions to local contexts and prioritize resources to areas at highest risk of poliovirus transmission [8,33]. It is also essential to assess the impact of disruptions like the COVID-19 pandemic on population immunity levels, particularly in regions already vulnerable to poliovirus circulation.

Serological surveys provide critical insights into population immunity levels by measuring the presence of neutralizing antibodies against each poliovirus serotype in the blood [35]. These surveys complement surveillance data on acute flaccid paralysis (AFP) cases and environmental sampling, offering a more direct measure of a population’s protection against poliovirus infection and disease [36]. In Pakistan, seroprevalence studies have been instrumental in identifying immunity gaps that may not be apparent from vaccination coverage data alone, as immunization effectiveness can be compromised by factors such as vaccine failure, improper vaccine handling, and interference from maternal antibodies or concurrent enteric infections [19].

This study presents the results of a series of four seroprevalence surveys conducted in polio high-risk areas in Pakistan from 2016 to 2023, targeting children between 6 to 11 months of age. The purpose of these surveys was to understand the change immunity level for all three poliovirus serotypes 1, 2, and 3 by geographical area of survey over the years.

By examining variations over time in seroprevalence for all poliovirus serotypes, this study presents insights into the effectiveness of vaccination strategies implemented, the impact of program disruptions by external challenges, such as the COVID-19 pandemic, and the emergence of immunity gaps that may require targeted interventions [37]. Additionally, comparative analysis facilitates the identification of variations in immunity levels that may not be apparent from single cross-sectional assessments [38]. The findings from the first three rounds of the survey have already been reported in other studies [39,40].

## 2. Materials and Methods

### 2.1. Study Design and Setting

This study employed a cross-sectional design, consisting of four consecutive rounds of population-based serological surveys conducted in polio high-risk areas of Pakistan from November 2016 to October 2023. These surveys were carried out by the Aga Khan University (AKU) in Karachi in close collaboration with the provincial and national emergency operations centers (NEOC) for polio eradication in Pakistan. The first round (R1) was conducted in 2016–2017, the second round (R2) in 2017, the third round (R3) in 2018–2019, and the fourth round (R4) in 2022–2023. The first three rounds of the surveys were conducted in 12 districts, and the fourth round included 44 districts. These districts were categorized as the high-risk region for polio by the NEOC. A list of the districts for each round of the survey is provided in Appendix A. These districts served as geographical strata for the surveys.

### 2.2. Sampling Procedure

The NEOC’s Lot Quality Assurance Sampling (LQAS) cluster database served as the sampling frame for each survey round. A two-stage cluster sampling technique was employed to select clusters from the geographical strata and eligible households from the selected clusters. In each geographical stratum, the polio program’s designated immunization zones were regarded as clusters. From each geographic stratum, 25 clusters were chosen using the probability proportional to size (PPS) method. Following the selection of the clusters, a household listing was conducted. Using this fresh household listing, the necessary number of households with children aged 6 to 11 months were chosen at random. Only one target child was chosen from each household. The Kish grid approach was used to choose one child from each household if there was more than one child in the target age group.

### 2.3. Sample Size Estimation

For the first three survey rounds, assuming a true seroprevalence of 95%, a design effect of 1.5, and a 90% individual response rate, the sample size was determined to estimate seroprevalence with a 95% confidence level and a ±5% margin of error. This resulted in a sample size of 294 children per stratum (almost 12 children per cluster), and a sample of 300 children from each stratum was covered. For the fourth round, a true Seroprevalence of 90% was assumed for sampling size estimation, while the other factors for sample size estimation remained the same as those of the first three rounds. This resulted in a required sample size of 210 children aged between 6–11 months from each stratum (9 children from each cluster).

### 2.4. Data Collection

Using a standardized questionnaire, the surveys gathered data on socioeconomic position, household demographics, and immunization history for every child participating in the study. The immunization data were recorded from vaccination cards where cards were available, and in their absence, data were collected through mother/caregiver recall. The questionnaire was designed to collect data from the mothers or caretakers of children enrolled in the study. It was translated into the national language and pilot-tested in the field with 100 participants from non-survey locations. In addition to completing the questionnaire, a qualified phlebotomist collected 2 mL of venous blood from every participating child. No replacements were made for households that refused participation or were locked.

Our survey team comprised data collectors, phlebotomists, team leaders, social mobilizers, regional supervisors, lab technicians, and medical officers. A single data collection team consisted of a male team leader, one male and one female social mobilizer, and two data collectors and two phlebotomists (all females). Team leaders and supervisors underwent a five-day central training conducted by experienced faculty from AKU. Afterward, cascade trainings were organized at the regional level for the data collection teams.

### 2.5. Laboratory Methodology

After collection, the blood sample was centrifuged to separate the serum, which was then transferred into sterile, labeled cryovials. The samples were stored in a cold box with ice packs and transported to the closest AKU laboratory collection outlet. From these collection points, the samples were sent to the AKU laboratory in Karachi. During transportation, the temperature of the samples was monitored using a thermometer attached to the ice box. At the AKU laboratory, two distinct aliquots were prepared: one was dispatched to the National Institute of Health (NIH), Pakistan, for neutralization assay testing, and the other was retained as a backup [41]. A poliovirus neutralizing antibody titer >1:8 was used to define seropositivity in this study [42].

### 2.6. Statistical Analysis

The primary outcome of this study was to assess the level of serological protection (seropositivity) against poliovirus types 1, 2, and 3 in children aged 6–11 months by survey round and geography. We performed descriptive analyses to summarize the characteristics of the study population. Proportions of seroprevalence were compared descriptively by survey rounds, provinces, and polio high-risk cities to assess variations in population immunity. No statistical trend analysis or modeling was conducted since each survey round was an independent cross-sectional assessment. Results were aggregated at the provincial level and presented as percentages with 95% CIs and graphically represented to show variations by survey rounds. STAT (version 18) was used for analyses [43].

## 3. Results

Throughout the four rounds of the survey, 68 polio high-risk districts were covered. Several districts were repeatedly surveyed, and additional districts were included in the fourth round after their reclassification as high-risk due to changing reports of polio cases. Altogether, 2084 clusters, 32,371 households, and 32,907 children aged 6–11 months were covered. The proportion of male children was slightly higher than that of female children in all rounds. Vaccination card retention was higher in the initial rounds; however, there was a substantial decline in the fourth round (40%) (Table 1). OPV coverage ranged between 68% and 75% while IPV coverage varied between 55% and 82% across the survey rounds. Over two-thirds of the caregivers were illiterate. Round-wise, district-specific demographic and coverage indicators are presented in the Appendix A.

### 3.1. Seroprevalence of Polio Type-1 Across Provinces and Survey Rounds

Poliovirus type 1 seroprevalence was consistently high in Punjab, Sindh, and KPK, with prevalence estimates over 95% in all four survey rounds. Immunity levels were between 98.3% and 99.7% in Punjab, and between 97.3% and 99.2% in Sindh. In KPK, it was consistently above 95%, though a minor drop was seen in the fourth round (95.8%). On the other hand, Balochistan showed more variability, the lowest coverage of 91.5% being reported in round 1 and another steep decline in round 4 (89.7%). These results reflect overall high immunity in Punjab, Sindh, and KPK, but persisting and high immunity gaps in Balochistan, especially in the latest round (Figure 1, Appendix A).

### 3.2. Seroprevalence of Polio Type-2 Across Provinces and Survey Rounds

The seroprevalence of poliovirus type 2 was found to be heterogeneous among provinces and survey rounds, with lower prevalences than type 1. Punjab had the highest prevalence in round 1 (83.4%, 95% CI: 79.2–87.7), followed by Sindh (61.7%, 95% CI: 59.4–64.1) and a comparatively lower level in Balochistan (59.8%, 95% CI: 56.7–63.0). In round 2, Punjab (55.3%, 95% CI: 49.6–61.0) and Sindh (52.9%, 95% CI: 50.4–55.3) showed a decline, whereas Balochistan (78.0%, 95% CI: 75.3–80.6) and KPK (67.7%, 95% CI: 65.1–70.3) recorded higher seroprevalence. Round 3 showed heterogeneity as Sindh continued to fall further to 34.6% (95% CI: 32.2–36.9), but Balochistan reached the highest level at 82.1% (95% CI: 79.6–84.5), while KPK had a prevalence of 78.1% (95% CI: 75.8–80.5). By round 4, every province displayed decreasing levels, with Punjab (53.2%, 95% CI: 51.3–55.1), Sindh (43.9%, 95% CI: 42.6–45.2), and KPK (57.4%, 95% CI: 56.3–58.5) showing medium prevalence, while the lowest seroprevalence was in Balochistan (40.4%, 95% CI: 38.9–41.8) (Figure 1, Appendix A).

Results revealed that type 2 immunity was always below type 1 in all provinces, with Sindh and Punjab showing the largest declines after the first round, whereas Balochistan generally exhibited variable but high seroprevalence until a steep decline in the fourth round.

### 3.3. Seroprevalence of Polio Type-3 Across Provinces and Survey Rounds

Seroprevalence of type 3 poliovirus was overall high in provinces and rounds, similar to type 1 but with greater provincial variation. Punjab reported the highest rates throughout, at nearly 99% in rounds 1 and 2. Sindh and KPK also had high rates (>94% for most rounds), although both fell to approximately 91% in round 4. Balochistan demonstrated considerably lower type 3 immunity, beginning at 86.5% (95% CI: 84.3–88.7) in round 1 and falling to 81.1% (95% CI: 79.9–82.2) in round 4 (Figure 1, Appendix A).

The results reflect that type 1 immunity across the country remained stable and high, type 2 immunity was stable yet lower with wide provincial variations, and type 3 immunity was moderately elevated but decreasing, where Balochistan lagged for all three serotypes consistently.

### 3.4. Seroprevalence of Polio Type-1 Across High-Risk Cities

Type 1 poliovirus seroprevalence was analyzed in high-risk cities across the survey rounds. Karachi and Peshawar demonstrated persistently high seroprevalence ranging between 97% and 99% throughout the study period. On the other hand, Quetta showed greater variability, with seroprevalence increasing from approximately 94.8% (CI:95%; 92.3–97.3) in round 1 to 99.7% (CI:95%; 99.0–100.3) in round 3, followed by a decline to 96.4% (CI:95%: 94.7–98.1) in round 4. In Killa Abdullah, the prevalence remained fluctuated over time, increasing from 87.8% (CI:95%; 84.2–91.5) in round 1 to 97.5% (CI:95%; 95.7–99.2) in round 3 before a steep decline to 76.2% (CI:95%; 72.2–80.1) in round 4. A similar pattern was observed with seroprevalence in Pishin, where it increased from 91.9% (CI:95%; 88.8–94.4) in round 1 to 98.3% (CI:95%; 96.99–99.8) in round 3, followed a decline to 91.7% (CI:95%; 89.2–94.3) in round 4 (Figure 2, Appendix A).

The prevalence rates indicate that while Karachi and Peshawar have maintained stable and high immunity against type 1 poliovirus, Quetta, Killa Abdullah, and Pishin experienced greater fluctuations and a notable decrease in seroprevalence in recent years.

### 3.5. Seroprevalence of Polio Type-2 Across High-Risk Cities

Type 2 poliovirus seroprevalence declined in the high-risk cities, with notable variability between the locations. In Karachi, the prevalence declined from 65.2% (CI:95%; 62.2–68.1) in round 1 to 38.5% (CI:95%; 35.4–41.6) in round 3, with a slight recovery to 46.8% (CI:95%; 45.1–48.5) in round 4. Peshawar exhibited an initial decline from 64.6% (CI:95%; 60.8–68.3) in round 1 to 57.9% (CI:95%; 54.1–61.7) in round 2, followed by a substantial improvement to 69.9% (CI:95%; 65.7–74.1) in round 4. In Quetta, the prevalence displayed variable patterns; increasing from 74.1% (CI:95%; 69.2–79) in round 1 to 93.1% (CI:95%; 90.2–95.9) in round 3, before it declined to 61.7% (CI:95%; 57.3–66.1) in round 4 (Figure 2, Appendix A). A similar pattern was observed in Killa Abdullah, maintaining levels between 61–66% in the first three rounds, before it declined to 36.7% (CI:95%; 32.3–41.2) in round 4. Similarly, in Pishin, a quite dramatic improvement was witnessed in the prevalence, increasing from 44.5% (CI:95%; 38.9–50) in round 1 to 88.66% (CI:95%; 85–92.2) in round 2 and maintaining it at 87.4% (CI:95%; 83.7–91.2) in round 3 before it experienced a steady decline in round 4 [41.3% (CI:95%; 36.7–45.9) (Figure 2, Appendix A).

These prevalence rates indicate substantial variability in type 2 poliovirus immunity across the high-risk cities, highlighting potential gaps in vaccination coverage that may require targeted strategies for specific regions.

### 3.6. Seroprevalence of Polio Type-3 Across High-Risk Cities

The variations in immunity for poliovirus type 3 were nearly identical to type 1, with seroprevalence being consistently high in most districts. Karachi and Peshawar remained well above 93% through all rounds, even peaking at over 97% in rounds 2 and 3. In Quetta, the type 3 prevalence improved from 92.9% (CI:95%; 90–95.8) in round 1 to 96.7% (CI:95%; 94.7–98.7) in round 3 before slightly declining to 89.8% (CI:95%; 87–92.5) in round 4. Similarly, in Pishin, the prevalence increased from 86.7% (CI:95%; 82.9–90.5) in round 1 to 97% (CI:95%; 95.1–98.9) in round 3 and then decreased to 84.6% (CI:95%; 81.3–87.9) in round 4. On the other hand, Killa Abdullah had persistently lagged behind other districts, with seroprevalence for type 3 ranging from 80.1% (CI:95%; 75.7–84.6) in round 1 to a peak of 94.3% (CI:95%; 91.7–96.8) in round 3, followed by a steep drop to 67.5% (CI:95%; 63.1–71.8) in round 4 (Figure 2, Appendix A).

These findings indicate that while most cities maintained a high level of immunity (above 90%) against type 3 poliovirus, Killa Abdullah showed a concerning situation, with a suboptimal seroprevalence level. The 95% confidence intervals suggest that these patterns are statistically significant, especially for the sharp decline in Killa Abdullah. Overall, these findings underscore the need for targeted interventions in Killa Abdullah and continued vigilance in other regions to sustain and strengthen immunity against type 3 poliovirus in these high-risk populations.

## 4. Discussion

This study assesses the immunity levels among children aged 6–11 months for all three strains of poliovirus in Pakistan’s high-risk region between 2016 and 2023. The results identify gaps in immunity and regional disparities influenced by national immunization efforts and operational challenges. The findings provide valuable insights into the effectiveness of current strategies for delivering immunization services in the region.

Throughout the study period, the consistently high seroprevalence for type 1 poliovirus (>90%) was observed in most areas, especially in Punjab and Sindh (98–99%), which was a major programmatic success. This stability in immunity levels reflects the continued focus on type 1 poliovirus in both routine immunization and supplementary vaccination campaigns since it is the only endemic wild poliovirus serotype still present in Pakistan [6,25,29]. However, the decline in poliovirus type 1 immunity in Balochistan to 89% in 2022–2023 reflects challenges of maintaining immunity level posed by security situations, remote geography, and health system constraints [29].

Among high-risk cities, the type 1 immunity in Karachi and Peshawar—historical poliovirus reservoirs—remained remarkably high and stable, demonstrating that intensive vaccination campaigns in these urban centers have been largely successful in building and maintaining population immunity [5,10]. Conversely, the decline in immunity in Quetta despite past improvements indicates vulnerability to immunity gaps, a concern supported by environmental surveillance findings that could potentially sustain virus transmission [22,28]. The overall stability of type 1 seroprevalence, even during periods of programmatic disruption due to the COVID-19 pandemic, suggests that prioritization of type 1 vaccination activities has been effective [25,26]. However, the decreasing pattern in several regions during the 2022–2023 survey indicates for continued vigilance, particularly in densely populated areas with inadequate sanitation infrastructure [44]. Similar patterns have been reported in Afghanistan, where continuous WPV1 transmission in border regions is associated with pockets of low immunity to type 1 poliovirus caused by insecurity, limited access, and difficult geographical terrains [33]. On the other hand, countries such as India have effectively eliminated WPV1 through the implementation of targeted national SIAs; a robust, regular vaccination program; and efficient surveillance [36,45]. These comparative experiences demonstrate the importance of health system strength, community trust, logistical reach, and robust monitoring to maintain high immunity and eradicate polioviruses.

Type 2 poliovirus immunity displayed a relatively high variation, caused by the 2016 global switch from tOPV to bOPV, which excluded vaccination for type 2 poliovirus [14,15]. As anticipated, nonetheless, concerning, this global switch led to declines in type 2 seroprevalence in Pakistan, with rates dropping below 40% in Balochistan and Sindh in 2022–2023. This declining pattern across the provinces leaves a substantial population vulnerable to type 2 poliovirus infection, including vaccine-derived strains [11,12], and also highlights a systemic challenge rather than region-specific issues, underscoring the impact of the global OPV switch on population immunity level [3].

Our findings reflect that among polio high-risk cities, type 2 immunity patterns varied greatly. In Karachi, the immunity level was low and continued to decline throughout the study period, whereas in Pishin, it remarkably improved (90%) and remained stable post-2017–2018, likely due to the implementation of SIAs with mOPV2 or IPV in response to cVDPV2 outbreaks [5,16]. Our findings related to type 2 immunity levels and patterns are aligned with the epidemiological reality of cVDPV2 outbreaks in Pakistan during the study period [12]. The persistent low immunity in multiple regions created conditions favorable for the emergence and circulation of vaccine-derived strains, complicating eradication efforts [13]. The partial recovery observed in some regions by 2022–2023 may reflect the impact of targeted type 2 vaccination responses, but the continued low levels in other areas suggest that maintaining adequate type 2 immunity remains a significant challenge in the post-switch period [16].

Similar to the findings of our study, post-switch type 2 immunity declines were reported in Nigeria and the Democratic Republic of the Congo, where cVDPV2 outbreaks occurred due to inadequate IPV coverage and delayed mOPV2 responses [3,13]. However, countries that swiftly implemented mOPV2 in their immunization campaigns, such as Benin and Egypt, demonstrated more promising control of cVDPV2 spread, highlighting the importance of a timely response for evolving issues around poliovirus circulation following the 2016 global switch [3,11]. These proven strategies reinforce the need for a strategic implementation of mOPV2 in Pakistan in areas with low type 2 immunity.

For type 3 poliovirus, immunity level generally improved over the study period in most regions, likely due to continued inclusion of bOPV containing type 1 and 3 strains in routine immunization and supplementary campaigns in the post-switch era [14]. However, by 2022–2023, several areas experienced a significant decline; for instance, in Killa Abdullah, type 3 immunity dropped from 94% in 2017–2018 to 69% in 2022–2023, creating a significant immunity gap that could potentially facilitate virus transmission if type 3 strains were introduced [46]. The timing of this reduction can be associated with the distortion in routine immunization activities in the wake of the COVID-19 pandemic, which could have influenced the immunity level in the population. This also highlights how high-risk communities are vulnerable to external programmatic shocks [25,26]. Aligned with our findings, studies from Yemen and Indonesia also recorded a decline in immunity level for type 3 poliovirus in the aftermath of type poliovirus eradication in 2019, mainly due to reduced focus on it [35,47,48]. These findings suggest that eradication certification must not lead to contentment in immunization efforts, especially in polio high-risk areas and under-immunized communities like Pakistan.

The pronounced regional disparities in immunity level for all three strains of polio viruses in our study mirror the underlying differences in access to immunization services, health system capacity, and operational reach [28]. Urban centers like Karachi maintained high type 1 and type 3 immunity and maintained consistency in type 2 protection level. Peripheries and border districts in Balochistan, like Killa Abdullah, demonstrated more volatile immunity patterns across all serotypes, likely due to challenges related to cross-border movement, accessibility, and security concerns [31,33,35,36].

These patterns underscore the need for granular and context-specific strategies in delivering immunization services. Though national-level coverage may appear adequate, gaps in immunity at the subnational level and certain pockets can reinforce virus circulation [49].

Although our study did not explicitly analyze the impact of COVID-19 disruptions, the declines in immunity level between 2018–2019 and 2022–2023 provide insights into potential consequences of the pandemic on immunity level for all three serotypes. These disruptions likely contributed to missed vaccinations and weakened the health system outreach, disproportionately impacting the already underserved communities [24,25,26,27].

The findings of this study highlight several policy considerations. Maintaining high immunity for type 1 should be the cornerstone of eradication efforts with robust monitoring in areas showing early signs of decline, and for type 3 immunity, the eradication certification should not derail the immunity maintenance, and efforts must be sustained to uphold the immunity level in under-immunized communities. For type 2 immunity gaps, demand integrated approaches, including administration of mOPV in SIAs and IPV in both routine and outreach sessions with enhanced surveillance. Monitoring of immunity level supported by environmental and AFP surveillance is needed for informed data-driven decision making and to close the immunity gaps contributing towards the ultimate goal of a polio-free Pakistan.

The study has several limitations. Since each survey round was cross-sectional in design and independently implemented, the study provides population-level immunity estimates but does not allow for assessing causal relationships or tracking individual-level changes over time. The vaccination history was in part captured through caregiver recall, in the absence of vaccination cards. This could result in the unintentional bias of recall about vaccination status. Even though this timeframe also included the COVID-19 epidemic, its particular effects on immunization activities and the subsequent immunity gaps were not specifically captured during this study. Thus, any interpretation with regard to the disruptions of the pandemic remains inferential. In spite of these limitations, our study provides variations and inequities in poliovirus immunity across the “at-risk” regions of Pakistan, highlighting the gaps that demand programmatic efforts.

## 5. Conclusions

This study assessed immunity levels against all three strains of poliovirus among 6–11-month-old children through four consecutive rounds of population-based cross-sectional serological surveys conducted in polio high-risk areas of Pakistan between 2016 and 2023. The findings from this series of serological evaluations indicate a promising yet variable progression in the developing and maintaining population immunity, with Baluchistan consistently reflecting lower seroprevalences for all three serotypes, particularly type 2. These results underscore the necessity for ongoing vigilance and the implementation of context-specific strategies to preserve the high type 1 immunity, improve and stabilize type 3 immunity, and address gaps in type 2 immunity through targeted interventions utilizing mOPV2 and IPV. Moreover, the establishment of robust surveillance mechanisms, tailored outreach initiatives directed at underserved populations, and improving immunization coverage are critical for closing immunity gaps and ultimately eradicating poliovirus in the country.

## Figures and Tables

**Figure 1 vaccines-13-01067-f001:**
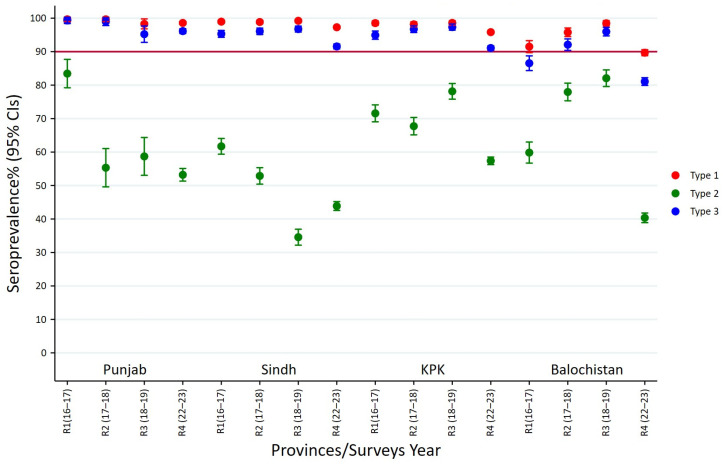
Polio seroprevalence by types across provinces and survey rounds.

**Figure 2 vaccines-13-01067-f002:**
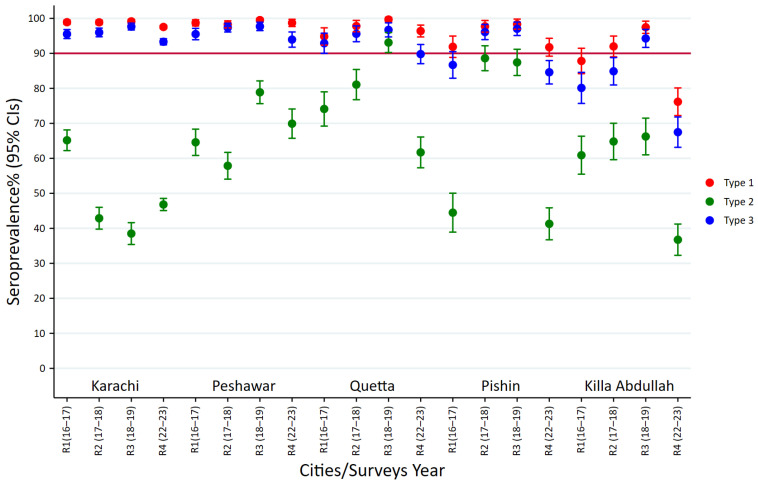
Polio seroprevalence by types across high-risk cities and survey rounds.

**Table 1 vaccines-13-01067-t001:** Survey round-wise demographic information and vaccination coverage.

Survey Rounds	Number of Districts	Clusters Covered	Households Covered	Number of Children Surveyed (N)	Male (%)	Vaccination Card (%)	OPV3 Coverage (%)	IPV Coverage (%)	Caregiver Education Level (Illiteracy Rate-%)
1	10	326	4143	4146	52.0	65.3	75.3	69.2	73.3
2	10	325	4093	4094	52.2	60.3	69.0	74.0	76.6
3	10	324	3985	3987	53.0	70.8	68.8	81.9	73.2
4	38	1109	20,150	20,680	51.9	40.0	74.1	55.4	67.5

Abbreviations: IPV: Inactivated Poliovirus Vaccine; OPV: Oral Poliovirus Vaccine.

## Data Availability

Data will be made available on request.

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
