# Peer review of "Seroprevalence of Poliovirus Types 1, 2, and 3 Among Children Aged 6–11 Months: Variations Across Survey Rounds in High-Risk Areas of Pakistan"

_vaccines, 2025, doi:10.3390/vaccines13101067_

Round 1

Reviewer 1 Report

Comments and Suggestions for Authors

Thank you for the opportunity to review this manuscript ID: vaccines-3933318.

Comments:

In the entire paper (from the title onwards), and especially in the objectives of the paper, the terms `trend` and `temporal trend` should be used cautiously, because neither the methodology described nor the presented results show how the trends were assessed and analyzed.

Lines 136-146: Specify the `Study design` applied in this manuscript.

Lines 198-207: Harmonize the statistical methodology with the `Study design` applied in this manuscript.

Line 217: Check whether the statement in the sentence `Over two-thirds of the caregivers were literate.` is in accordance with the data shown on the Table 1.  

A limitation of the study cannot be what was not foreseen by this study. Namely, on Lines 417-419 it is stated `Although data were collected through multiple rounds from 2016 to 2023, the survey design was cross-sectional, which limits the ability to draw causal inferences about changes in immunity at the individual level.`, since in the Materials and Methods section it is stated that the data on the seroprevalence of poliovirus types 1, 2, and 3 in this manuscript refer to `children 6-11 months`.   

Author Response

To the Editors,

Thank you for sharing the reviewers’ comments and providing us with the opportunity to submit a revised draft of the manuscript titled “Seroprevalence of poliovirus types 1, 2, and 3 among children aged 6-11 months: Cross-year trends in high-risk areas of Pakistan– Vaccines-3933318”. We appreciate the time and effort that you and the reviewers have dedicated to providing your valuable feedback. Those comments are all valuable and very helpful for revising and improving our paper. We have studied the comments carefully and have been able to incorporate changes to reflect most of the suggestions provided by the reviewers. Here is a point-by-point response to the reviewers’ comments.

Reviewer 1

Comments

Responses

In the entire paper (from the title onwards), and especially in the objectives of the paper, the terms `trend` and `temporal trend` should be used cautiously, because neither the methodology described nor the presented results show how the trends were assessed and analyzed.

We appreciate the reviewer’s insightful observation. We acknowledge that the study design and analyses were descriptive, comparing immunity levels across four survey rounds rather than performing formal statistical trend analyses. Accordingly, we have revised the wording throughout the manuscript to more accurately reflect this.

The title has been updated to:
“Seroprevalence of poliovirus types 1, 2, and 3 among children aged 6–11 months: Variations across survey rounds in high-risk areas of Pakistan.”

Lines 136-146: Specify the `Study design` applied in this manuscript.

We thank the reviewer for this comment. We have now specified the study design in the Methods section. The manuscript has been revised to indicate that this study design was a cross-sectional design [138–139].

Lines 198-207: Harmonize the statistical methodology with the `Study design` applied in this manuscript.

We appreciate the reviewer’s observation. The Statistical Analysis section has been revised to ensure consistency with the cross-sectional study design.

Line 217: Check whether the statement in the sentence `Over two-thirds of the caregivers were literate.` is in accordance with the data shown on the Table 1.  

We thank the reviewer for pointing this out. We have cross-checked the data presented in Table 1 and revised the text to ensure consistency between the narrative and the table values.

A limitation of the study cannot be what was not foreseen by this study. Namely, on Lines 417-419 it is stated `Although data were collected through multiple rounds from 2016 to 2023, the survey design was cross-sectional, which limits the ability to draw causal inferences about changes in immunity at the individual level.`, since in the Materials and Methods section it is stated that the data on the seroprevalence of poliovirus types 1, 2, and 3 in this manuscript refer to `children 6-11 months`

We thank the reviewer for this observation. We agree that the inability to assess causal relationships or individual-level changes is inherent to the cross-sectional design rather than an unforeseen limitation. The sentence in the Limitations section has been revised to clarify this distinction.

Reviewer 2 Report

Comments and Suggestions for Authors

The manuscript is devoted to evaluating immune protection against all three serotypes of poliovirus among children aged 6–11 months in high-risk polio-transmission zones in Pakistan, the reseach is relevant because poliomyelitis remains a serious public health concern, particularly in countries with unstable healthcare systems and insufficient vaccination rates. Assessing vaccine effectiveness and herd immunity is crucial for strategic decision-making to prevent outbreaks of the disease. There is an urgent need to identify vulnerabilities and adjust prevention strategies accordingly. Determining the level of protection in children aged 6–11 months is essential for planning further vaccinations and maintaining high collective immunity.

The presented research fills an important gap in the topic, contributes to raising awareness about children's immune status, improving vaccination programs, and mitigating risks of polio epidemics in the region.

The article provides unique data on the dynamics of polio immunity among children in Pakistan, one of the two remaining countries worldwide where wild-type poliovirus type 1 endemicity persists. The results of four rounds of surveys reveal long-term trends and regional differences in protective levels, allowing vulnerable population groups and regions requiring enhanced immunization efforts to be identified.

Investigation of the transition from trivalent oral polio vaccine (tOPV) to bivalent OPV (bOPV) following the withdrawal of type 2 strain in April 2016 sheds light on how vaccination policies affect population-level immunity and highlight deficiencies needing correction. Scientists can refine vaccination campaign plans to maintain high herd immunity.

Significant differences in antibody levels against different poliovirus serotypes (types 1, 2, and 3) are noted between provinces and cities, emphasizing the heterogeneity of vaccination coverage and the importance of targeted interventions. For example, Baluchistan province regularly demonstrated much lower immunity to all serotypes, necessitating special attention and intensified efforts in this region.

The study reveals that traditional vaccination coverage indicators are inadequate for fully grasping actual population protection. Serologic surveillance enables medical professionals to assess true immune defense levels and respond quickly to emerging pockets of low immunity, preventing disease outbreaks.

The material carries significant implications, informing strategies for polio eradication and shaping national and international vaccination policy.

Discussion of data is logical, tables and figures are informative; methods are relevant. The manuscript can be published after minor revision.

1) Expanding conclusion part would be beneficial; a brief summary of performed research could help the understandability of material.

2) Doi are provided only for a part of references.

Author Response

Reviewer 2

Comments

Responses

The manuscript is devoted to evaluating immune protection against all three serotypes of poliovirus among children aged 6–11 months in high-risk polio-transmission zones in Pakistan, the research is relevant because poliomyelitis remains a serious public health concern, particularly in countries with unstable healthcare systems and insufficient vaccination rates. Assessing vaccine effectiveness and herd immunity is crucial for strategic decision-making to prevent outbreaks of the disease. There is an urgent need to identify vulnerabilities and adjust prevention strategies accordingly. Determining the level of protection in children aged 6–11 months is essential for planning further vaccinations and maintaining high collective immunity.

We thank the reviewer for acknowledging the relevance and importance of this study.

The presented research fills an important gap in the topic, contributes to raising awareness about children's immune status, improving vaccination programs, and mitigating risks of polio epidemics in the region.

We thank the reviewer for their positive feedback and recognition of this study’s contribution.

The article provides unique data on the dynamics of polio immunity among children in Pakistan, one of the two remaining countries worldwide where wild-type poliovirus type 1 endemicity persists. The results of four rounds of surveys reveal long-term trends and regional differences in protective levels, allowing vulnerable population groups and regions requiring enhanced immunization efforts to be identified.

We thank the reviewer for highlighting the novelty and significance of our findings.

Investigation of the transition from trivalent oral polio vaccine (tOPV) to bivalent OPV (bOPV) following the withdrawal of type 2 strain in April 2016 sheds light on how vaccination policies affect population-level immunity and highlight deficiencies needing correction. Scientists can refine vaccination campaign plans to maintain high herd immunity.

We thank the reviewer for this valuable observation and fully agree that the transition from trivalent OPV (tOPV) to bivalent OPV (bOPV) provides important insights into how vaccination policy changes influence population-level immunity. This aspect has been discussed in the manuscript, where we highlight that the decline in type 2 immunity observed after the global withdrawal of the type 2 strain reflects the consequences of this policy shift.

Significant differences in antibody levels against different poliovirus serotypes (types 1, 2, and 3) are noted between provinces and cities, emphasizing the heterogeneity of vaccination coverage and the importance of targeted interventions. For example, Baluchistan province regularly demonstrated much lower immunity to all serotypes, necessitating special attention and intensified efforts in this region.

We thank the reviewer for highlighting this important point. We agree that the observed differences in seroprevalence across provinces, particularly the consistently lower immunity levels in Balochistan, underscore the heterogeneity of vaccination coverage and the need for targeted interventions. This aspect has been discussed in the manuscript, where we emphasize the importance of focused strategies and intensified immunization efforts in regions demonstrating persistently low population immunity.

The study reveals that traditional vaccination coverage indicators are inadequate for fully grasping actual population protection. Serologic surveillance enables medical professionals to assess true immune defense levels and respond quickly to emerging pockets of low immunity, preventing disease outbreaks.

We thank the reviewer for emphasizing this important point. We fully agree that traditional vaccination coverage indicators may not accurately reflect true population-level protection, and that serological surveillance provides a more precise measure of immunity. This point has been discussed in the manuscript, where we highlight that repeated seroprevalence surveys enable the identification of immunity gaps that may not be apparent through administrative coverage data alone, thereby supporting timely and targeted immunization responses to prevent outbreaks.

The material carries significant implications, informing strategies for polio eradication and shaping national and international vaccination policy.

We thank the reviewer for recognizing the policy relevance of this study. We agree that the findings have important implications for shaping national and global polio eradication strategies. This aspect has been highlighted in the Conclusion, where we note that the results can inform targeted vaccination planning and contribute to policy decisions aimed at sustaining population immunity and achieving eradication goals.

Discussion of data is logical, tables and figures are informative; methods are relevant. The manuscript can be published after minor revision.

We sincerely thank the reviewer for their positive feedback and appreciation of the study’s design, data presentation, and analysis. We are grateful for the recommendation for publication following minor revisions and have carefully addressed all comments and suggestions to further strengthen the manuscript.

1) Expanding conclusion part would be beneficial; a brief summary of performed research could help the understandability of material.

We thank the reviewer for this valuable suggestion. In response, we have expanded the Conclusion section to include a concise summary of the study objectives, main findings, and their implications for vaccination policy and eradication strategies.

2) Doi are provided only for a part of references.

We thank the reviewer for pointing this out. We have carefully reviewed the reference list and added missing DOIs for all references where they are available.

We tried our best to improve the manuscript by incorporating the feedback from the reviewers. We hope that the correction will meet with approval.

Once again, thank you very much for your comments and suggestions.

Regards,

Dr. Sajid Bashir Soofi

Professor, Department of Pediatrics & Child Health,

Director, Centre of Excellence in Women and Child Health

The Aga Khan University, Stadium Road, Karachi - 74800, Pakistan,

sajid.soofi@aku.edu, +92-21-34864798.

October 13, 2025

Round 2

Reviewer 1 Report

Comments and Suggestions for Authors

Thank you for the opportunity to re-review manuscript ID: vaccines-3933318.

The authors correctly addressed all my comments point by point, and made appropriate corrections in the revised version of this manuscript.
I thank the authors for their efforts in revising this manuscript.

Good job!